# Evidence of pandemic fatigue associated with stricter tiered COVID-19 restrictions

**Federico Delussu, Michele Tizzoni**[ID][☯], **Laetitia Gauvin**[ID][☯] *

ISI Foundation, via Chisola 5, 10126, Turin, Italy

☯ These authors contributed equally to this work.
* laetitia.gauvin@isi.it

**Data Availability Statement:** Movement data used in this study are publicly available at: https://data.humdata.org/dataset/movement-range-maps and https://www.google.com/covid19/mobility/. All

## Abstract

Despite the availability of effective vaccines against SARS-CoV-2, non-pharmaceutical interventions remain an important part of the effort to reduce viral circulation caused by emerging variants with the capability of evading vaccine-induced immunity. With the aim of striking a balance between effective mitigation and long-term sustainability, several governments worldwide have adopted systems of tiered interventions, of increasing stringency, that are calibrated according to periodic risk assessments. A key challenge remains in quantifying temporal changes in adherence to interventions, which can decrease over time due to pandemic fatigue, under such kind of multilevel strategies. Here, we examine whether there was a reduction in adherence to tiered restrictions that were imposed in Italy from November 2020 through May 2021, and in particular we assess whether temporal trends in adherence depended on the intensity of the restrictions adopted. We analyzed daily changes in movements and in residential time, combining mobility data with the restriction tier enforced in the Italian regions. Through mixed-effects regression models, we identified a general trend of reduction in adherence and an additional effect of faster waning associated with the most stringent tier. We estimated both effects being of the same order of magnitude, suggesting that adherence decreased twice as fast during the strictest tier as in the least stringent one. Our results provide a quantitative measure of behavioral responses to tiered interventions—a metric of pandemic fatigue—that can be integrated into mathematical models to evaluate future epidemic scenarios.

## Author summary

Pandemic fatigue, the decreased motivation to adhere to social distancing measures and adopt health-protective behaviors, represents a significant concern for policymakers and health officials, as novel SARS-CoV-2 variants undermine the effects of vaccinations and non-pharmaceutical interventions become measures of last resort. Here, we investigate the effects of pandemic fatigue by measuring the temporal variation in adherence to mobility restrictions in Italy, during the period November 2020—May 2021, when a tiered restriction system was adopted to mitigate the spread of COVID-19. We measure such effect by analyzing large-scale mobility traces from Google and Facebook, in the Italian

other data is provided as a Supplementary File to the manuscript.

**Funding:** The study was partially supported by the Lagrange Project of the ISI Foundation funded by the CRT Foundation. The funders had no role in study design, data collection and analysis, decision to publish, or preparation of the manuscript.

**Competing interests:** The authors have declared that no competing interests exist.

regions, through a statistical model that accounts for a global temporal trend and a local time trend, associated with the stringency of the interventions. Our results show that adherence to the restrictions decreased over time, and that it decreased faster when the strictest tier, the red one, was in place. Our study provides evidence that adherence to interventions can wane at different pace, depending on their stringency: such insight can help evaluating the interplay between mobility restrictions, behavior and disease dynamics in epidemic models.

## Introduction

Since the beginning of November 2020, to fight COVID-19 Italy adopted a system of tiered social distancing measures based on 3 levels of increasingly stricter restrictions coded as yellow, orange, and red [1]. The system has remained in place since then, on a regional basis, and the activation of a specific tier is automatically enforced according to a weekly epidemiological risk assessment that takes into account healthcare capacity and local COVID-19 incidence. Although vaccination uptake has reached more than 80% of the eligible population in Italy, and booster doses have been administered since early October 2021, non-pharmaceutical interventions (NPIs) may still be needed to curb the rise of infections during winter months. Wider circulation of SARS-CoV-2 and the emergence of new variants of concern may require the enforcement of increasingly higher restriction tiers but a reduced motivation to comply, and lowered individual risk perception may decrease the adherence to a new wave of social distancing policies.

Temporal variations in adherence to protective behaviors against COVID-19 were observed during the first pandemic wave in different countries and they were characterized as a possible consequence of *pandemic fatigue* [2]. The concept of pandemic fatigue is loosely defined as a decreased motivation to adopt health-protective measures, and its existence has been hypothesized since the early phases of the COVID-19 pandemic [3]. According to the WHO, pandemic fatigue is the *demotivation to follow recommended protective behaviours, emerging gradually over time and affected by a number of emotions, experiences and perceptions* [4]. There has been substantial debate about the existence and quantifiability of such phenomenon, which has been harshly criticized when it was invoked as an argument against mitigation policies in the UK [5, 6]. However, previous studies have shown that individual willingness to comply with protective behaviors changed over time, and varied across kinds of behaviors and across personal psychological traits [7, 8]. During the first COVID-19 wave, reductions in adherence were found to be stronger for high-cost behaviors, such as physical distancing, while low-cost behaviors, such as mask adoption, did not change substantially from initial levels [2]. After the initial adoption of blanket lockdowns of January-March 2020, to face the following infection waves a few governments worldwide have resorted to staged restriction systems, where social distancing measures are calibrated according to local risk factors [9]. For instance, this happened in the UK [10], Italy [1], China [11] and California [12]. From a policy perspective, it is important to assess whether the previously observed effects of pandemic fatigue are also present and measurable when a tiered restriction system is in place. In principle, a more gradual tightening of NPIs may support higher levels of compliance, however, whether temporal trends in adherence are expected to remain constant for any degree of restrictions remains an open question. Evaluating the long-term sustainability of tiered NPIs, together with their effectiveness against the spread of SARS-CoV-2 [13, 14], is key to mount an effective response to the epidemic.

In this study, we investigated how adherence to mobility restrictions changed in time under a tiered system, over the course of 7 months in Italy, through the analysis of mobility indicators publicly available from Facebook and Google. Italy represents a peculiar case, since NPIs have been structured into a tiered system for more than one year and the system has been consistently enforced in all regions, in face of two consecutive waves of COVID-19 resurgence during the the Fall/Winter 2020–2021. By analyzing mobility data in the 20 Italian regions, we explored how changes in adherence to the restrictions varied over time, and whether such temporal variations were different by color tier. The main goal of our study was measuring behavioral patterns that could potentially hint at a reduced adherence, as an indicator of pandemic fatigue, and relate such patterns to the restriction tier in place.

## Results

We gathered the daily relative change of individual mobility as measured by Facebook and Google (see Methods for more details on these indicators), in each of the 20 Italian regions, and we associated the relative change in mobility with the restriction tier that was in place each day. Fig 1 displays the daily percentage change in movement by Facebook users in the Italian regions between November 2020 and June 2021, where each data point is coloured according to the corresponding tier: yellow, orange or red. Data from Google of the same period are shown in S1 Fig. The two indicators capture two different and complementary aspects of mobility: while Facebook measures the change in the number of movements, Google estimates the change in time spent at home, thus providing an indicator of stillness.

As it can be seen from Fig 1, every region experienced all the three restriction tiers in different non-consecutive periods. Each tier introduced specific mandates to limit individual movements and increase social distancing, ranging from a 10 PM—5 AM curfew in the yellow tier, up to a general stay-at-home mandate in the red tier. In the Methods, we provide a complete description of the measures introduced by each tier.

We characterized the trends in adherence to social distancing by adopting mixed-effects regression models to analyze the temporal evolution of the change in movement, or the change in time spent at home. To assess the presence of both a general time trend—that is, a global effect of fatigue—and a local time trend—that is a faster or slower change in adherence in presence of a specific tier—we compared the results of 5 different model specifications (see the Methods for a full definition). The most detailed model includes both a general time trend and a local time trend associated with each color tier, and a random intercept for each region and for each tier. The coefficient of the general time trend measures the overall trend in movement since the beginning of the restrictions, that is from November 6, 2020. The coefficients associated to the local time variable measure the trend of change in movement since the date of introduction of a new tier. To evaluate the relevance of a model that includes both a global time trend and a local time trend dependent on the tier, we introduced 3 simpler models and one model that controls for risk perception. The 3 simpler models are: one that includes only the global time trend, one that considers only the local time trend, independent of the tier, and finally a model that includes only a local time trend dependent on the tier. To control for risk perception, we add to the full model one epidemiological covariate to be used as a control variable. We consider three different variables to this end: the daily reported number of positive cases by region, the daily number of new hospitalizations and daily ICU admissions.

Tables 1 and 2 report the estimated coefficients for the first 4 models considered, for the relative change in movement and the change in residential time, respectively. Overall, we consistently found a global trend of increase in mobility (decrease in time spent at home, respectively) both when including or excluding the local time trends ($\gamma_{1,0}$ of Models 1 and 4 in

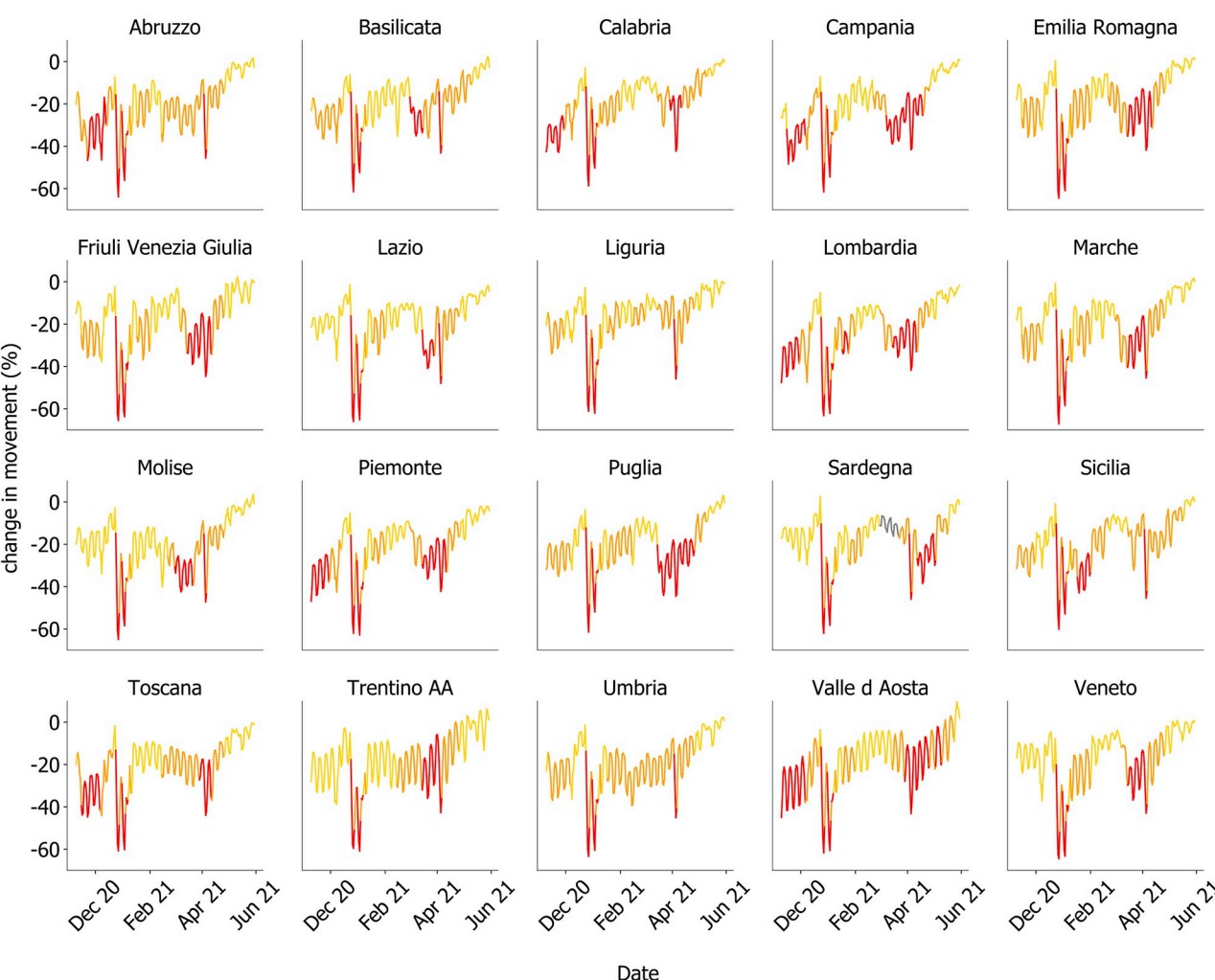

**Fig 1. Mobility changes and tiered restrictions.** Daily relative change of mobility—as measured by Facebook—with respect to the baseline in the Italian regions, ordered alphabetically from top to bottom. Color coding (yellow, orange, red) indicates the tier that was in place on a given day. Gray indicates the absence of tiered restrictions (only in Sardegna). For a detailed definition of this metric see the Methods section.

Tables 1 and 2). The two models performed best, when considering their fit to the data both in terms of their adjusted $R^2$ and Akaike Information Criterion (AIC) values. This shows that change in movements effectively increased over the full period of study, an effect that can be interpreted as a general decrease in adherence to the restrictions. Models that included a local time effect only, either independent of the color tier (Model 2) or different by tier (Model 3), were characterized by a lower statistical performance. However, the inclusion of a different local trend by color was favored with respect to an equal effect by tier, as indicated by the difference of AIC, $\Delta_{AIC} = 6$ for Facebook data and $\Delta_{AIC} = 20$ for Google data. This indicates that the local trend was dependent on the color tier that was put in place. As can been seen from Tables 1 and 2, the full model (Model 4) that included both the global time trend and the local time trend by color tier, provided the best fit to the data (adj-$R^2 = 0.454$ for Facebook data, adj-$R^2 = 0.707$ for Google data) and it was the most likely according the AIC value.

Fig 2 displays the estimated coefficients' values, and their associated error, using the relative change in movement as dependent variable of the full model. Overall, results suggest that

**Table 1. Model results for the relative change in movement as dependent variable.** Estimates of the regression coefficients and their standard error (in parenthesis). Each column corresponds to a different model.

| | *Dependent variable*: | | | |
| --- | --- | --- | --- | --- |
| | Change in movement (%) | | | |
| | **(1)** | **(2)** | **(3)** | **(4)** |
| Global time trend | | | | |
| $\gamma_{1,0}$ | 0.082*** | | | 0.082*** |
| | (0.003) | | | (0.003) |
| Local time trend | | | | |
| $\gamma_{2,0}$ | | 0.116*** | 0.255*** | 0.155*** |
| | | (0.017) | (0.050) | (0.045) |
| $\gamma_{2,1}$ (orange) | | | −0.183*** | −0.149*** |
| | | | (0.058) | (0.052) |
| $\gamma_{2,1}$ (yellow) | | | −0.135** | −0.165*** |
| | | | (0.057) | (0.051) |
| Intercept | | | | |
| $\gamma_{0,0}$ | −39.339*** | −32.202*** | −33.626*** | −41.047*** |
| | (0.862) | (0.936) | (1.073) | (0.993) |
| $\gamma_{0,1}$ (orange) | 9.598*** | 9.695*** | 11.768*** | 11.343*** |
| | (0.501) | (0.561) | (0.862) | (0.770) |
| $\gamma_{0,1}$ (yellow) | 18.024*** | 18.949*** | 20.487*** | 19.990*** |
| | (0.489) | (0.547) | (0.890) | (0.794) |
| Observations | 3,222 | 3,222 | 3,222 | 3,222 |
| Adjusted $R^2$ | 0.453 | 0.313 | 0.315 | 0.454 |
| AIC | 23,638 | 24,368 | 24,362 | 23,632 |

*Note*:

*p<0.1;

**p<0.05;

***p<0.01

adherence to the measures decreased over time, as indicated by a significant and positive trend of the mobility change during the study period (Fig 2, gray). Model's coefficients estimated on the variation in residential time measured by Google provided a similar picture, as shown in Fig 3. Such effect was estimated to be equal to 0.08% daily increase in the relative change of movements and 0.04% increase in time spent outside home (which we interpret as the opposite of the change in residential time, see caption of Fig 3).

Furthermore, we found an additional effect that suggests a faster decline in adherence associated with the red tier. Indeed, after an introduction of the red tier, and before changing to a different color, individual mobility increased faster than the general time trend, with an additional increase in the relative mobility of 0.16% per day (Fig 2, red) and an additional decrease in residential time of 0.04% per day (Fig 3, red). On the other hand, movements during the orange and yellow tiers did not display a significant association with an additional decrease or increase in adherence (Fig 2) with respect to the general trend. When considering the residential data, the model did not find any statistically significant effect of the orange tier while each introduction of a yellow tier was characterized by a small yet significant reduction of the time spent at home, equal to a 0.02% daily decrease. To better contextualize our results, the estimated general time trend corresponds to more than 15% increase in the relative mobility change over the whole study period. Moreover, while 2 weeks under a yellow tier would lead

**Table 2. Model results for the relative change in residential time as dependent variable.** Estimates of the regression coefficients and their standard error (in parenthesis). Each column corresponds to a different model.

| | Dependent variable: | | | |
| --- | --- | --- | --- | --- |
| | Change in residential time (%) | | | |
| | (1) | (2) | (3) | (4) |
| Global time trend | | | | |
| $\gamma_{1,0}$ | −0.040*** | | | −0.039*** |
| | (0.001) | | | (0.001) |
| Local time trend | | | | |
| $\gamma_{2,0}$ | | −0.057*** | −0.092*** | −0.045*** |
| | | (0.005) | (0.016) | (0.011) |
| $\gamma_{2,1}$ (orange) | | | 0.066*** | 0.051*** |
| | | | (0.018) | (0.013) |
| $\gamma_{2,1}$ (yellow) | | | 0.017 | 0.028** |
| | | | (0.018) | (0.013) |
| Intercept | | | | |
| $\gamma_{0,0}$ | 18.814*** | 15.312*** | 15.592*** | 19.225*** |
| | (0.218) | (0.293) | (0.335) | (0.250) |
| $\gamma_{0,1}$ (orange) | −4.056*** | −4.087*** | −4.810*** | −4.628*** |
| | (0.125) | (0.173) | (0.267) | (0.193) |
| $\gamma_{0,1}$ (yellow) | −7.248*** | −7.554*** | −7.676*** | −7.520*** |
| | (0.121) | (0.169) | (0.274) | (0.198) |
| Observations | 3,401 | 3,401 | 3,401 | 3,401 |
| Adjusted $R^2$ | 0.705 | 0.433 | 0.436 | 0.707 |
| AIC | 15,645 | 17,873 | 17,853 | 15,626 |

*Note*:

*p<0.1;

**p<0.05;

***p<0.01

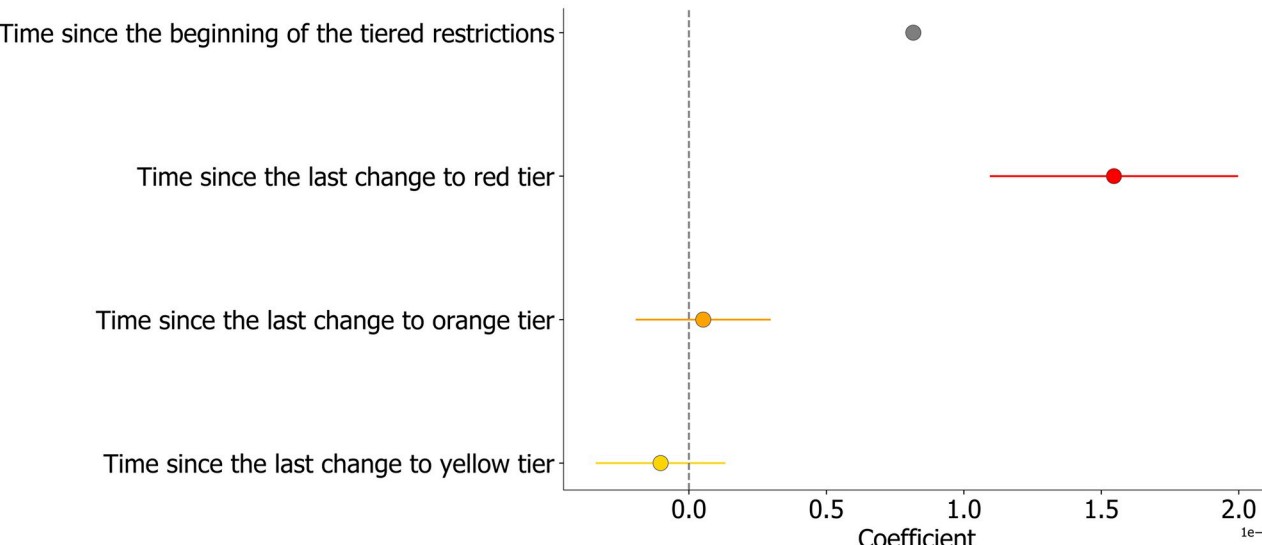

**Fig 2. Model estimates for the relative change in mobility.** Estimates of the regression coefficients of the mixed-effects model when the daily relative change in mobility is the dependent variable. Point estimates are evaluated from Table 1. Error bars correspond to the Standard Error, colors indicate the associated tier.

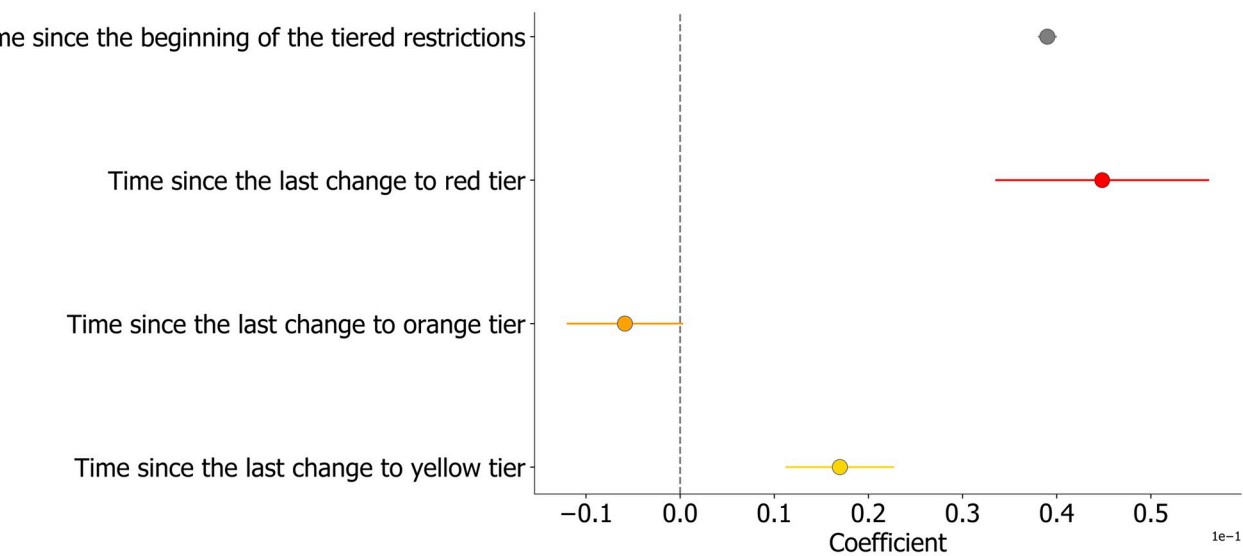

**Fig 3. Model estimates for the relative change in time spent outside home.** Estimates of the regression coefficients of the mixed-effects model when the daily change in residential time is the dependent variable. Point estimates are evaluated from Table 2. Error bars correspond to the Standard Error, colors indicate the associated tier. To ease a direct comparison with the results obtained with Facebook data, we reverted the sign of the dependent variable, so that it can be interpreted as the relative change in time spent outside home.

to an average increase in movements of about 1%, 2 weeks under the red tier would lead to an average 3% increase in the relative mobility. Overall, the strictest tier led to a faster increase in the relative mobility and a faster decrease in the time spent at home.

Including an epidemiological covariate as a control variable (Model 5), marginally improved the model fit but it did not affect the results on the changes in adherence as shown in S3 and S4 Tables. The results were robust in all cases, either considering regional COVID-19 incidence, or hospitalizations, or ICU admissions, as a proxy for risk perception.

## Discussion

In our study, we evaluated temporal trends in the adherence to mobility restrictions, under a tiered system. Our model highlighted the presence of a temporal variation in adherence to mobility restrictions, which can be broadly ascribed to the effect of pandemic fatigue [2]. In particular, our statistical analysis showed that changes in adherence were faster during periods characterized by the strictest level of restrictions (the red tier). Interestingly, the magnitude of the two effects—the decrease in adherence over time and specifically during the red tier—was of a similar order. This means, in practice, that the introduction of the red tier doubled the speed of reduction in adherence with respect to the adoption of the yellow tier only. The analysis of two complementary data sets led to the same result, although the effect measured with residential data was smaller than the one observed with movement data. This is not surprising, because variations in the daily time spent at home can not exceed the natural limit of 50% and they are usually much smaller. Indeed, we expect to see variations in the effect we measured, depending on the choice of the target mobility metric. Recently, Weill and collaborators [15] have shown that the choice of mobility measures can lead to different outcomes in studies aimed at assessing the impact of NPIs on human movements. It would be interesting to extend our work by investigating changes in adherence through other indicators that can be derived from mobile phone data, such as spatial proximity or the radius gyration [16, 17].

In a recent paper, analyzing 6 different mobility metrics from Italy over the period September 25—November 25, 2020, Manica and coauthors noticed a possible reduction in compliance during the red tier [1]. However, due to the short time frame under study, they could not derive any conclusion on this effect. Our study confirmed their early remark, through a robust statistical analysis, by considering the lift and the introduction of different tiers over a much longer time frame of 7 months.

Our results have important implications for modelling efforts aimed at assessing future epidemic scenarios under the adoption of different NPIs and vaccination strategies. A few authors incorporated changes in adherence in their modeling studies so far [18–21], while models usually assumed a sustained adherence over time due to the lack of available data. According to the results of our analysis, when performing scenario analysis, epidemic modelers should incorporate a faster reduction in adherence in presence of stricter restrictions. It is important to note that our work does not allow to draw conclusions regarding the effectiveness of a specific set of NPIs against the spread of SARS-CoV-2. Despite the observed reduction in adherence over time, and its additional effects with and increased restriction tier, high intensity interventions, introduced for a relatively short period of time, may still be the best policy option to drastically curb the epidemic spread. A recent work by Di Domenico et al. [19] based on French data found that the loss of adherence occurred faster during the second lockdown than the first one. However, even when taking this effect into account, the authors have shown that milder interventions tend to be sub-optimal in the long run, due to the higher transmissibility of new variants and the sustained pressure on the healthcare system that inevitably follows large viral circulation.

Our work comes with a set of limitations. First, we derived our conclusions from observational data that are subject to biases. Mobility data from both Facebook and Google are collected from users who opted-in to share their location history and therefore may not be fully representative of the whole population. However, both data sources achieve a high spatial coverage in Italy, as discussed in previous studies [22, 23] and they have been extensively used to assess the impact of NPIs during the pandemic, worldwide [24–27].

Another source of uncertainty is the observational nature of our study. Despite the consistent reduction in adherence we measured using two independent datasets, we can only speculate about the underlying causes. Additional research based on individual survey data is needed to untangle the psychological determinants of health-related behaviors and in particular of pandemic fatigue [28–30]. Our analysis allowed us to measure changes in behaviors related to movements, however it does not provide insights into other types of behaviors, such as face masks wearing, hand hygiene or reduction in close-proximity contacts, which could exhibit different and even increasing adherence patterns [2]. Mobility, however, remains a reliable proxy to measure community transmission, in particular during periods of sustained viral circulation [26, 31, 32]. In our model we took into account regional differences by including a varying intercept, however it is important to observe that spatial heterogeneities in responses to restrictions can be partially explained by socio-economic determinants, such as the local structure of the labour force [33, 34] or wealth disparities [35]. Finally, the time frame of our study includes two important events, that could have affected the adherence to restrictions with two opposite effects. On the one hand, the beginning of the mass vaccination campaign in Italy, which may have affected the adherence to restrictions by inducing changes in risk perception [36]. The vaccination campaign in Italy officially started on December 27, 2020, however, it was not until April 2021 that vaccine administrations took off, reaching about 30% of the whole population by May 15 [37]. On the other hand, in the same period, a highly pathogenic pandemic wave caused by the Alpha variant hit the country [38], thus potentially leading to increase the adoption of health protecting behaviors. Untangling the combined effects of a

high vaccination coverage and the emergence of new, more transmissible, variants on the behavioral responses to NPIs remains an open research question that will be important to address in future work.

In conclusion, we have shown that in a system of tiered restrictions, adherence can be difficult to sustain over time and more so when the most stringent measures are enforced. We focused on the specific case of Italy, due to the availability of relatively long time series, but our approach can be easily extended to other countries, in presence of a similar tiered intervention system. As NPIs remain an important tool against COVID-19 even with widespread vaccination coverage [39], our results will be useful to inform epidemic models in the design of optimal intervention policies for future pandemic waves.

## Materials and methods

### Mobility data

To evaluate time trends in mobility changes, we used two publicly available data sources. The first is the Movement Range Maps released by the Facebook Data for Good program and made publicly available through the Humanitarian Data Exchange platform at [40]. Our main quantity of interest is the Change in Movement metric, during the period starting from November 6, 2020—when the tiered restrictions were first adopted—until May 30, 2021 that is the last day of 2021 before at least one region removed all restrictions (entering the so-called white tier). This mobility metric relies on the number of 16-level Bing tiles (whose size is approximately 600×600 meters at the Equator) that are visited every day by Facebook users in the 20 Italian regions. The Change in Movement is then defined as the relative change in the average number of tiles visited by the users of a given region with respect to a baseline that pre-dates the introduction of social distancing measures. Additional details on the metric definition can be found at [41].

The second data source are the COVID-19 Community Mobility Reports made publicly available by Google [42]. Since March 2020, Google provides daily time-series of mobility changes across different categories of places, such as retail and recreation, groceries and pharmacies, parks, transit, residential. Data is collected, aggregated and anonymized with differential privacy, from Android users who have opted-in to Location History [25, 43]. Mobility changes are computed with respect to a baseline that corresponds to mobility levels observed before the pandemic, between January and February 2020. In this study, our main quantity of interest is the relative mobility change in Residential category, defined as the relative change in total time spent by users in residential areas.

In the following, we will denote the relative change in movement, independently from the data source, as $m_{r,t}$, where $r$ stands for region and $t$ for time, in days. However, it is important to note that the two quantities of interest are expected to vary along different directions, as a consequence of social distancing: the Change in Movement will decrease, taking negative percentage values, and the Residential movement will increase, taking positive percentage values.

### Tiered restrictions

The system of three-tiers restrictions has been introduced in Italy on November 6, 2020. Each tier is associated with a color, according to increasing levels of restrictions: yellow, orange, red. Tiers are assigned on a regional basis, according to the epidemiological situation and, in particular, to a risk assessment based on several indicator, including the weekly COVID-19 incidence, the time-varying reproductive number and the proportion of occupied ICU beds, among others.

Independently from the tier, the following measures were always in place during the study period (November 6, 2020—May 30, 2021):

- Mandatory face mask wearing in outdoor spaces.

- Closure of all indoor recreational, sport and cultural venues.

- 50% capacity reduction of all public transport services.

  Moreover, the **yellow tier** introduced the additional measures:

- Stay-at-home mandate between 10pm and 5am (except for work, health and other certified reasons).

- Shopping malls closed during weekends and holidays (with the exception of essential retail & services).

- Distance learning in high schools and universities.

- No service in cafes, bars and restaurants after 6pm and take away allowed until 10pm.

  The **orange tier** introduces stricter measures as follows:

- Stay-at-home mandate between 10pm and 5am and ban on movements between municipalities and to/from other regions (except for work, health and other certified reasons).

- Shopping malls closed during weekends and holidays (with the exception of essential retail & services).

- Distance learning in high schools and universities.

- Closure of all cafes, bars and restaurants. Take away allowed until 10pm.

  Social distancing measures in **red tier** were:

- Stay-at-home mandate and ban on movements between municipalities and to/from other regions (except for work, health and other certified reasons).

- Closure of all shops (with the exception of essential retail & services).

- Distance learning in second and third grade of middle schools, in all grades of high schools and universities.

- Closure of all cafes, bars and restaurants. Take away allowed until 10pm.

## Statistical model

To estimate time trends in the evolution of the movement relative change across regions, we run a time series regression with mixed-effects, with $m_{r,t}$ as the dependent variable. To measure the time trend both globally, i.e. over the period of tiered restrictions, and locally, i.e. over each new period of a given tier, we introduced two independent variables: the time since the beginning of the tiered restrictions, $t$, and the time since the last change of tier $\Delta t$. To control for level of perceived risk associated with the infection, we included epidemiological covariates, $e(r, t)$, at regional level. Such covariates are: the daily number of confirmed COVID-19 cases, the daily number of hospitalizations, the daily number of ICU admissions. The general equation for the models we fit is given by:

$$m_{r,t} = \beta_{0,r}(color) + \beta_1 * time + \beta_{2,t}(color) * \Delta t + \beta_3 * e(r, t) \tag{1}$$

where:

$$\beta_{0,r} = \gamma_{0,0} + \gamma_{0,1}(color) + \gamma_{0,2}(region)$$
$$\beta_1 = \gamma_{1,0}$$
$$\beta_{2,t} = \gamma_{2,0} + \gamma_{2,1}(color) \qquad (2)$$
$$\beta_3 = \gamma_{3,0}$$

We compare the results of different model specifications, by fitting 5 versions of the model defined above:

1. the model only considers the global time trend, estimating the intercepts $\beta_{0,r}(color)$ and $\beta_1$;

2. the model only considers the local trend, i.e. the trend after each change of tier independently on the tier color, by setting all the coefficients equal to 0 except for $\beta_{0,r}(color)$ and $\gamma_{2,0}$;

3. we consider the local trend, taking in consideration the color, looking for $\beta_{0,r}(color)$ and $\beta_{2,t}(color)$;

4. we consider both the global and the local trends, measuring $\beta_{0,r}$, $\beta_1$, and $\beta_{2,t}$.

5. we consider both the global and the local trends, and we control for risk perception through $\beta_3$, searching for all the coefficients.

We used the function $lm()$ of the R $stats$ package to fit the 5 models [44].

In our analysis, we excluded data from the period from December 15, 2020 to January 10, 2021 as most of those days were calendar holidays that strongly affected individual mobility, an effect that could bias the final results. Sardinia is the only region that experienced a short period of white tier (indicating no restrictions) in March, much earlier than all other regions, and for this reason we excluded Sardinia from the analysis. Moreover, in the case of Trentino Alto Adige, the spatial area considered by Facebook to compute the mobility changes and the definition of region for the tier assignment do not match. For this reason we excluded Trentino Alto Adige from the analysis of Facebook data. Overall, both regions represent less than 3% of the Italian population.

We present the results of the first 4 models in Tables 1 and 2, for the two datasets under study. We provide the intercept values for the regions in S1 and S2 Tables of the Supporting Information.

Results of model 5, evaluated with 3 different epidemiological covariates, are presented in S3 and S4 Tables.

The coefficient values have to be interpreted with respect to a reference value. In our case, the Abruzzo region and red were taken as references for region and tier color, respectively. To be more precise, the value of $\gamma_{0,0}$ corresponds the mean movement relative change during the red zone in the region of Abruzzo. The same coefficient for another region is obtained by summing $\gamma_{0,0}$ and the intercept value of that region given in S1 or S2 Tables. The slope of the local trend during a period of orange (resp. yellow) tier (independently on the region) is given by the sum of $\gamma_{2,0}$ and $\gamma_{2,1}(orange)$ (respectively, $\gamma_{2,1}(yellow)$). Figs 2 and 3 report the total effects, that is the sums of the estimated coefficients and their reference for an ease of interpretation.

## Supporting information

**S1 Fig. Changes in residential mobility and tiered restrictions.** Daily relative change in the residential time—as measured by Google—with respect to baseline, in the Italian regions

ordered alphabetically from top to bottom. Color coding indicates the tier that was in place each day. Gray indicates the absence of restrictions.
(PDF)

**S1 Table. Model intercepts by region for the relative change in movement as dependent variable.**
(PDF)

**S2 Table. Model intercepts by region for the relative change in residential time as dependent variable.**
(PDF)

**S3 Table. Model results for the relative change in movement as dependent variable, controlling for risk perception.**
(PDF)

**S4 Table. Model results for the relative change in residential time as dependent variable, controlling for risk perception.**
(PDF)

**S1 Data. Daily values of COVID-19 incidence, hospitalization, ICU admission and tier by region.**
(CSV)

## Acknowledgments

We gratefully acknowledge Lorenzo Ruffino for his help collecting data about the timeline of restrictions.

## Author Contributions

**Conceptualization:** Michele Tizzoni, Laetitia Gauvin.

**Data curation:** Federico Delussu, Michele Tizzoni, Laetitia Gauvin.

**Formal analysis:** Michele Tizzoni, Laetitia Gauvin.

**Investigation:** Michele Tizzoni, Laetitia Gauvin.

**Methodology:** Michele Tizzoni, Laetitia Gauvin.

**Supervision:** Michele Tizzoni, Laetitia Gauvin.

**Validation:** Michele Tizzoni, Laetitia Gauvin.

**Visualization:** Michele Tizzoni, Laetitia Gauvin.

**Writing – original draft:** Michele Tizzoni, Laetitia Gauvin.

**Writing – review & editing:** Federico Delussu, Michele Tizzoni, Laetitia Gauvin.

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
