## [Decision Letter · Decision Letter 0]

15 Mar 2022

PDIG-D-21-00142

Evidence of pandemic fatigue associated with stricter tiered COVID-19 restrictions

PLOS Digital Health

Dear Dr. Gauvin,

Thank you for submitting your manuscript to PLOS Digital Health. After careful consideration, we feel that it has merit but does not fully meet PLOS Digital Health's publication criteria as it currently stands. Therefore, we invite you to submit a revised version of the manuscript that addresses the points raised during the review process.

We look forward to receiving your revised manuscript.

Kind regards,

Dylan A Mordaunt, MB ChB, MPH, MHLM, FRACP, FAIDH

Academic Editor

PLOS Digital Health

Journal Requirements:

1. Please amend your detailed Financial Disclosure statement. This is published with the article, therefore should be completed in full sentences and contain the exact wording you wish to be published.

State what role the funders took in the study. If the funders had no role in your study, please state: “The funders had no role in study design, data collection and analysis, decision to publish, or preparation of the manuscript.”

2. Please update your Competing Interests statement. If you have no competing interests to declare, please state: “The authors have declared that no competing interests exist.”

3. Please provide a complete Data Availability Statement in the submission form, ensuring you include all necessary access information or a reason for why you are unable to make your data freely accessible. Note that it is not acceptable for the authors to be the sole named individuals responsible for ensuring data access.

PLOS defines a study's minimal data set as the underlying data used to reach the conclusions drawn in the manuscript and any additional data required to replicate the reported study findings in their entirety. Any potentially identifying patient information must be fully anonymized. 

If your research concerns only data provided within your submission, please write “All data are in the manuscript and/or supporting information files.” as your Data Availability Statement.

4. We ask that a manuscript source file is provided at Revision. Please upload your manuscript file as a .doc, .docx, .rtf or .tex. If you are providing a .tex file, please upload it under the item type ‘LaTeX Source File’ and leave your .pdf version as the item type ‘Manuscript’.

5. Please provide separate figure files in .tif or .eps format only, and remove any figures embedded in your manuscript file. If you are using LaTeX, you do not need to remove embedded figures.

Please also ensure that all files are under our size limit of 20MB.

For more information about how to convert your figure files please see our guidelines: https://journals.plos.org/digitalhealth/s/figures

Additional Editor Comments (if provided):

Thank you for your submission. We received reviews from both epidemiologists and an economist. Reviewer 1 summarises this very well, the underlying question being whether the effectiveness of public health restrictions has a temporal dimension. The study is original in approach and relevant to the digital health community for the combination of public health analytics and the novel use of public data sources. With respect to the criteria for publication:

1) The study appears to be original work.

2) This issue is of high importance, with broad interest to the community of researchers, engineers and clinicians working in the field of digital health and indeed population health, economics and the broader social sciences.

3) The study demonstrates high methodological rigor and ethical standards. Reviewer 1 raises some issues that should be addressed/responded to, though some of these may appropriately be addressed in post-publication review. I will leave it to the authors to respond as they feel appropriate.

4) Substantial evidence for its conclusions.

5) Clearly outlined utility and accessibility for the broader community.

6) Follow appropriate standards and practice of open science.

Reviewers' comments:

Reviewer's Responses to Questions

**Comments to the Author**

1. Does this manuscript meet PLOS Digital Health’s publication criteria? Is the manuscript technically sound, and do the data support the conclusions? The manuscript must describe methodologically and ethically rigorous research with conclusions that are appropriately drawn based on the data presented.

Reviewer #1: Partly

Reviewer #2: Yes

2. Has the statistical analysis been performed appropriately and rigorously?

Reviewer #1: Yes

Reviewer #2: Yes

Reviewer #3: Yes

3. Have the authors made all data underlying the findings in their manuscript fully available (please refer to the Data Availability Statement at the start of the manuscript PDF file)?

Reviewer #1: Yes

Reviewer #2: Yes

Reviewer #3: Yes

4. Is the manuscript presented in an intelligible fashion and written in standard English?

Reviewer #1: Yes

Reviewer #2: Yes

Reviewer #3: Yes

5. Review Comments to the Author

Reviewer #1: I like this paper. It asks a critically important question: does compliance with public health measures wane with time and with the stringency of the measures. If it does, then public health measures need to take account of waning. If it doesn't, then officials who sometimes hold fire for fear of waning effects could be more confident in setting restrictions where appropriate.

The authors use daily mobility as measured by Facebook and Google as measure of compliance with public health orders during different tiers of restrictions.

I will refrain from going too deeply into the finer points on statistical methods - my concern is a bit broader.

Mobility will depend not only on the level of restrictions in place but also on how dangerous things look outside. Imagine that the governments implemented zero public health restrictions. If there were a lot of Covid around, a lot of people would just decide to stay home all on their own, even without being ordered to do so. And once things looked safe again, they'd start going outside again. So the mobility data would show a pattern that varied over time with the level of Covid in the community. It's why economist Josh Gans has argued that, over the longer term, R0 tends to 1: people go out when it feels safe, increasing R past one. Then it starts looking dangerous so they stay home, and R declines to below one, making it look safer again.

Covid restrictions are themselves endogenous to actual Covid levels. So when things start looking dangerous, the restrictions come in. When they start looking safe again, restrictions ease. But restrictions are a discrete value and the amount of Covid in the community is continuous. People will have *started* taking their own protective measures ahead of the restrictions, and will have *started* taking fewer precautions before the restrictions are eased. On the downside of that curve, are we seeing waning compliance with the measure because they're sick of the measures? Or are we seeing waning compliance because people are judging the world to look less risky and that, in their view, the measures are now less warranted? It will be a mix of the two. Disentangling it requires having some measure of the perceived riskiness of the environment. 

I think the statistical model needs to incorporate actual Covid prevalence in each place over time as a control - unless actual Covid levels in each region were the same at each point in time and only the restrictions varied. If that's the case, then regional variation in restrictions could identify. But I rather strongly suspect that regional restrictiveness will depend on regional Covid levels. 

If measures like number of positive tests or proportion of positive tests aren't available at the finer-grained local level, the local hospital burden could be a pretty good proxy. If the hospitals are overflowing, I'm staying home regardless of restrictions. Once the hospitals are running normally again, I'll be less inclined to comply with restrictions - nothing to do with frustration about restrictions, just that "Don't swim in the river" signs are superfluous when the river is a raging torrent, but are likely to be ignored on a hot day by a calm and beautiful swimming spot. Changing levels of compliance with the "Do Not Swim" sign in that case have nothing to do with waning support for the sign's restrictions as people become frustrated with it and everything to do with people's own assessment of whether the sign's warning is reasonable in the circumstances.

The second concern probably cannot easily be addressed but should be listed as a potential limitation of the work. There was a thoroughly depressing paper published in the NBER working paper series in December on the use of mobility data in exactly this kind of work. See Weill, J, M Stigler, O Deschenes and M Springborn. 2021. "Researchers' Degrees-of-Flexibility and the credibility of Difference-in-Differences Estimates: Evidence from the Pandemic Policy Evaluations", NBER Working Paper 29550. https://www.nber.org/papers/w29550 . They show that the American literature on the effects of lockdowns on mobility is, in short, a mess. There are tons of ways of measuring mobility, and they don't correlate well with each other. They point to this as introducing huge researcher degrees of freedom: basically, motivated researchers can pick the result they want by choosing the mobility measure that gives them that result. I have absolutely no reason to expect that this has happened in this paper. But it points to a danger in this work: none of the mobility measures are themselves very good if the choice of measure winds up substantially influencing whether restrictions work or not. To my mind, that argues for using a broad set of mobility measures, running a principal components analysis through it, seeing whether there's some underlying measure on which they all load, and use that factor as the measure of mobility. That would be a ridiculous amount of work here, and I do not know whether those additional measures are even available for Italy. But readers should be given a pretty strong warning that, in the American literature, choice of mobility measure can prove strongly influential on the results and that the work should be replicated using other mobility measures to check consistency. I also note that the NBER work uses a much broader array of controls looking at effects on mobility: precipitation, snowfall, mean temperatures. The paper is really worth going through closely.

Reviewer #2: I appreciate the authors for the analysis performed and the manuscript is well written. 

The statistical methodology used in this study are appropriate.

The results are found to be satisfactory.

I would recommend to check for the grammatical errors and rephrasing few sentences.

Reviewer #3: The study “Evidence of pandemic fatigue associated with stricter tiered COVID-19 restrictions” uses mobility data to parameterize adherence to tiered interventions over time in Italy accounting for social distancing measures that were in place in every region each day. Results of the study are of interest to public health and can be integrated into mathematical models to evaluate optimal intervention policies for future epidemics. Methods are sound, results adequately described and limitations acknowledged.

Tables and figures should be self-explanatory. Please explain coefficients’ labels in the first column of tables 1 and 2.

6. PLOS authors have the option to publish the peer review history of their article (what does this mean?). If published, this will include your full peer review and any attached files.

**Do you want your identity to be public for this peer review?** For information about this choice, including consent withdrawal, please see our Privacy Policy.

Reviewer #1: Yes: Eric Crampton

Reviewer #2: No

Reviewer #3: No

**Comments to the Author**

1. Does this manuscript meet PLOS Digital Health’s publication criteria? Is the manuscript technically sound, and do the data support the conclusions? The manuscript must describe methodologically and ethically rigorous research with conclusions that are appropriately drawn based on the data presented.

Reviewer #3: Yes

---

## [Editor Report · Decision Letter 1]

4 Apr 2022

Evidence of pandemic fatigue associated with stricter tiered COVID-19 restrictions

PDIG-D-21-00142R1

Dear Dr. Gauvin,

We are pleased to inform you that your manuscript 'Evidence of pandemic fatigue associated with stricter tiered COVID-19 restrictions' has been provisionally accepted for publication in PLOS Digital Health.

Best regards,

Dylan A Mordaunt

Academic Editor

PLOS Digital Health

Thank you for your submission, this now meets the criteria for publication.